# Design and Characterization of a Novel Venetoclax-Zanubrutinib Nano-Combination for Enhancing Leukemic Cell Uptake and Long-Acting Plasma Exposure

**DOI:** 10.3390/pharmaceutics15031016

**Published:** 2023-03-22

**Authors:** James Griffin, Yan Wu, Qingxin Mu, Xinyan Li, Rodney J. Y. Ho

**Affiliations:** 1Departments of Pharmaceutics, University of Washington, Seattle, WA 98195, USA; 2Departments of Bioengineering, University of Washington, Seattle, WA 98195, USA; 3School of Pharmacy, China Pharmaceutical University, Nanjing 210009, China

**Keywords:** leukemia, nanoparticles, combination therapy

## Abstract

Leukemia remains incurable partly due to difficulties in reaching and maintaining therapeutic drug concentrations in the target tissues and cells. Next-generation drugs targeted to multiple cell checkpoints, including the orally active venetoclax (Bcl-2 target) and zanubrutinib (BTK target), are effective and have improved safety and tolerability compared to conventional, nontargeted chemotherapies. However, dosing with a single agent frequently leads to drug resistance; asynchronous coverage due to the peak-and-trough time-course of two or more oral drugs has prevented drug combinations from simultaneously knocking out the respective drugs’ targets for sustained leukemia suppression. Higher doses of the drugs may potentially overcome asynchronous drug exposure in leukemic cells by saturating target occupancy, but higher doses often cause dose-limiting toxicities. To synchronize multiple drug target knockout, we have developed and characterized a drug combination nanoparticle (DcNP), which enables the transformation of two short-acting, orally active leukemic drugs, venetoclax and zanubrutinib, into long-acting nanoformulations (VZ-DCNPs). VZ-DCNPs exhibit synchronized and enhanced cell uptake and plasma exposure of both venetoclax and zanubrutinib. Both drugs are stabilized by lipid excipients to produce the VZ-DcNP nanoparticulate (d ~ 40 nm) product in suspension. The VZ-DcNP formulation has enhanced uptake of the two drugs (VZ) in immortalized leukemic cells (HL-60), threefold over that of its free drug counterpart. Additionally, drug-target selectivity of VZ was noted with MOLT-4 and K562 cells that overexpress each target. When given subcutaneously to mice, the half-lives of venetoclax and zanubrutinib were extended by approximately 43- and 5-fold, respectively, compared to an equivalent free VZ. Collectively, these data suggest that VZ in VZ-DcNP warrant consideration for preclinical and clinical development as a synchronized and long-acting drug-combination for the treatment of leukemia.

## 1. Introduction

Roughly 500,000 cases of leukemia are diagnosed worldwide every year, with about 300,000 patients succumbing to the disease [1]. Leukemia is a general group of blood cancers derived from malignant bone marrow cells. Historically, treatments for leukemia have not been able to truly cure the disease due to the cancer’s persistence in other systems within the body, including the lymphatic system and lymph nodes, to which cancer drugs may have limited access [2]. Molecularly targeted small molecule and antibody-based drugs have been developed to eliminate malignant cells. However, their cellular and cancer-laden tissue distribution and retention can be limited due to high clearance and tissue barriers [3]. Only a fraction of orally administered small molecule drugs are able to reach the leukemic cell target as significant percentages of such drugs are subject to metabolic or excretory elimination.

Chemotherapeutic agents, including chlorambucil (alkylating agent), fludarabine (purine analogue), and cyclophosphamide (alkylating agent), are effective treatments for leukemia. However, at therapeutic doses these chemotherapeutic agents may carry significant side effects that can limit their application, particularly in weaker or older patients [4]. With the introduction of molecularly targeted agents for specific druggable proteins that are overexpressed in leukemia, an additional safety margin is added. Newer clinical treatments for B-cell leukemia targeted to molecular checkpoints can be divided into three groups of compounds that inhibit the uncontrolled growth of B cells: (1) Bcl-2, a mitochondrial antiapoptotic protein, (2) Bruton’s tyrosine kinase (BTK) inhibitors (TKI’s), (3) monoclonal antibodies, several of which are targeted to CD20, a surface antigen on B cells [5]. As molecularly targeted agents, these three compounds reduce off-target toxicities compared to earlier drugs, making them a preferred treatment option for patients from a wide demographic range [6]. Biologic drugs have been largely successful, though their inherent structure limits them to only binding a single target per drug molecule. Resistance events to molecularly targeted single agent treatments are well documented in chronic use. Therefore, combination regimens (with multiple drugs targeted to varying mechanisms of action) are often used to reduce the risk of single-drug resistance [7]. Combination regimens may provide synergy derived from two or more drug substances that both inhibit multiple pathways and improve potency [8]. While the newer molecularly targeted drugs are typically approved as monotherapies, combination therapies with these newer drugs are also being considered for treatment durability.

Zanubrutinib is a second-generation TKI of Bruton’s Tyrosine Kinase (BTK) that has been recently introduced and approved by the FDA for several B cell-based blood cancers, including mantle cell lymphoma (MCL). The BTK inhibitor zanubrutinib is currently administered daily in an oral dosage form (considered an attractive treatment for patients). Due to a short plasma half-life of 2–4 h, oral zanubrutinib is administered twice daily to maintain adequate plasma concentrations of the drug [9]. As chronic twice-daily oral dosing may cause pill fatigue, patient compliance is often an issue due to the physically taxing nature of such chemotherapeutics [10]. Zanubrutinib is currently approved for patients with refracted MCL or similar diseases only as a monotherapy treatment, but combination therapy regimens consisting partly of zanubrutinib have been explored in a prominent phase 3 clinical trial, the SEQUOIA study [11]. This study, currently in progress, is focused on chronic lymphocytic leukemia (CLL) treatment; it has so far reported improved progression-free survival in patients receiving zanubrutinib monotherapy compared to bendamustine-rituximab, a commonly used treatment targeting CD20-positive cells. An additional arm of the study is exploring patient tolerance of zanubrutinib in conjunction with venetoclax, a small-molecule inhibitor of Bcl-2; results have been positive, with 50/51 patients responding to treatment, but the study is ongoing [12]. Studies have also indicated that the zanubrutinib-venetoclax combination can be used on leukemias beyond CLL.

As previously noted, even if oral venetoclax and zanubrutinib can be administered together, the asynchronized peak and time course of the two drugs will not provide consistent, sustained intracellular levels to maximally suppress leukemic cell growth. The percentage of oral drugs absorbed into blood from the gut mucosa is typically lower than that of intravenous injections. Drug metabolic enzymes found in the gut and liver may also reduce the percentage of active drug in the blood, leading to limited drug bioavailability, and, in some cases, sub-therapeutic plasma and intracellular dug levels. As a result, these events may lead to an increased risk of inducing drug resistant cells in tumor sites [13]. In addition, daily (or more frequent) dosing, typically necessary for oral dosage form, can be cumbersome for patients, as high local concentration in the gut after oral dosing may lead to gastrointestinal injury. Additionally, zanubrutinib is typically administered twice daily, which over time can lead to pill fatigue in patients, further limiting the treatment due to missed doses. To address these limitations, we have evaluated the feasibility of a drug delivery system in which lower but sustained therapeutic levels persist in the blood for an extended period of time through the development of a combination of drugs that are targeted to multiple proteins in the leukemic cells. Drug combinations composed of molecularly targeted drug substances could greatly improve both the potency and patient tolerance of the combination drug product.

We have previously demonstrated that DcNP can stabilize combinations of hydrophobic (lopinavir and ritonavir, LogP = 5.9 and 6, respectively) and hydrophilic (tenofovir and emtricitabine, LogP = −1.6 and −0.6, respectively) HIV drugs with amphipathic lipid excipients [14]. When given subcutaneously to nonhuman primates, DcNP both extends the plasma time course (long-acting behavior) of all three HIV drugs and leads to higher drug levels in lymphocytes than in plasma (demonstrating preferential cell targeting effects) [15].

In this study, we evaluated whether a venetoclax and zanubrutinib drug combination could be assembled into a similar DcNP lipid nanoparticle to provide both long-acting plasma exposure and adequate drug concentrations; in addition, we evaluated if enhanced, synchronized cell uptake could be achieved. We have found that a nanoparticle formulation can greatly extend the half-lives of certain drugs when administered subcutaneously in mice compared to equivalent free drugs or DcNPs administered intravenously. As their protein targets are expressed across multiple forms of leukemia, both venetoclax and zanubrutinib have shown efficacy against different types of leukemia in phase III clinical trials [11,12,13]. Thus, a DcNP approach for the venetoclax-zanubrutinib combination manifests broad treatment potential for several different types of leukemia beyond CLL.

## 2. Materials and Methods

### 2.1. Reagents

N-(carbonylmethoxypolyethyleneglycol-2000)-1,2-distearoyl-sn-glycero-3-phosphoethanolamine, sodium salt (DSPE-mPEG_2000_), and 1,2-Distearoyl-sn-glycero-3-phosphocholine (DSPC) (GMP grade) were purchased from Corden Pharma (Liestal, Switzerland). The two drug substances, zanubrutinib (BGB 3111) and venetoclax (ABT 199) were supplied by MedChemExpress (Monmouth Junction, NJ, USA). All other chemicals and solutions were purchased from Sigma-Aldrich (St. Louis, MO, USA) unless otherwise noted.

### 2.2. Preparation and Characterization of Drug Combination Nanoparticles

To prepare the drug combination containing venetoclax and zanubrutinib, 9.6 mg venetoclax and 9.6 mg zanubrutinib, plus 33.6 mg DSPE-mPEG_2000_ and 85.4 mg DSPC, were dissolved together in 1 mL organic solvent in a glass tube. For the first production attempt, the chemical components were dissolved in an organic solution of ethanol with 5% ammonia, and were then subjected to rotary evaporation, followed by reconstitution in 0.9% NaCl and 20 mM NaHCO_3_ buffer using a stir bar. An Avanti Polar Lipids sonicator (Avanti, Alabaster, AL, USA) was used to reduce particle size after reconstitution in aqueous solvent. Sonication was performed at 40–45 °C for 5 min, followed by 5 min rest, followed by a final 5 min of sonication. The suspended drug combination was stabilized by lipid excipients and referred to as the drug combination nanoparticle, or VZ-DcNP. This suspension was diluted to defined concentrations with buffer solution. The second production method used *tert*-butyl alcohol (TBA) as an organic solvent, which was then removed via rotary evaporation and an additional 4 h of lyophilization to remove residual TBA solvent. The diameter of particles in suspension was estimated with a NICOMP 380 ZLS (NICOMP, Chicago, IL, USA). The final production method, consisting of solvent rotary evaporation, lyophilization, and sonication, was selected for assessment both in vitro and in vivo. 

To estimate the percentage of venetoclax and zanubrutinib association to DcNP, the VZ-DcNP in suspension was first dialyzed (6–8 kDa molecular weight cutoff) in buffer under sink conditions. The sink conditions were generated by dialyzing 200 µL of VZ-DcNP suspension in 200 mL buffer solution (1000-fold volume change) for 4 h. The VZ-DcNP drug association efficiency (AE%) was determined by comparing the pre- and post-dialysis drug concentration ratios of each drug (V and Z). To determine VZ either in the dialysate or retentate, the samples were first extracted with organic solvent and analyzed with an LC-MS/MS assay as described below. Nanoparticles were also visually assessed for consistency using transmission electron microscopy with negative staining; sample suspensions containing the VZ-DcNPs were placed onto a TEM grid (copper, 300-mesh, coated with carbon and Formvar film), allowed to settle for 5 min, and then stained with 5% uranyl acetate as a negative stain. A Tecnai G2 F20 electron microscope (FEI, Hillsboro, OR, USA) was used at 200 kV.

### 2.3. Drug Extraction from VZ-DcNPs and LC-MS/MS Analysis

An extraction protocol was established to quantify concentrations of venetoclax and zanubrutinib in both nanoparticle-bound and free forms. Briefly, the drugs were solubilized by diluting the sample with ethyl acetate, which extracted them from either the DcNP complex, mouse plasma, or both. Following centrifugation, the supernatants were dried with nitrogen gas and then reconstituted in acetonitrile. Extracted drug solutions were then loaded onto a Shimadzu HPLC system coupled to a 3200 QTRAP mass spectrometer (Applied Biosystems, Grand Island, NY, USA). The HPLC system consisted of two Shimadzu LC-20A pumps, a DGU-20A5 degasser, and a Shimadzu SIL-20 AC HT autosampler. A Synergi Polar-RP column (100 × 2.0 mm) with a C_8_ guard column (4.0 × 2.0 mm) (Phenomenex, Torrance, CA, USA) was used for separations. Mobile phase A used water with 20 mM ammonium acetate and B used acetonitrile. The separations were done at room temperature with a flow rate of 0.55 mL/min. The mass spectrometer was equipped with an electrospray ionization (ESI) TurboIonSpray source, and the system was operated using Analyst software, version 1.5.2 (ABSciex, Framingham, MA, USA). Drug concentrations in various samples were calculated with standard curves prepared from normal mouse plasma containing known drug concentrations.

### 2.4. Drug Potency against Cancer Cells 

K-562, a human chronic myelogenous leukemia cell line, was purchased from ATCC (Manassas, VA, USA). Human leukemic cell line MOLT-4 (of acute lymphoblastic leukemia origin), and acute promyelocytic leukemia HL-60 cells were obtained from Carrie Cummings at Fred Hutchinson Cancer Research Center (Seattle, WA, USA). They were grown in RPMI medium 1640, which contained 1% 100× Antibiotic-Antimycotic (Thermo Fisher Scientific, Waltham, MA, USA) and 10% fetal bovine serum. These were selected for evaluation due to their different degree of drug target expression, specifically Bruton’s tyrosine kinase (BTK) and B-cell lymphoma 2 (Bcl-2). HL-60 cells express both BTK [16] and Bcl-2 [17], while K-562 [18,19] cells only express BTK and MOLT-4 [20,21] cells express only Bcl-2.

Each cell line was first allowed to grow in black 96-well assay plates (Corning, NY, USA). After 1 h, 200 µL of varying concentrations of each drug (venetoclax or zanubrutinib), a combination of both free drugs (*w/w* 1:1), the same combination in DCNPs (VZ-DCNPs), or no drug (medium control) were added in RPMI medium to the cells. On day 5, drug treatment effects were estimated using an AlamarBlue Cell Viability Assay (Thermo Fisher Scientific, Waltham, MA, USA). The viable cells that produced positive fluorescence signals were quantified with a PerkinElmer 1420 Multilabel Counter plate reader. AlamarBlue was diluted 10-fold with cell media; the existing cell culture media in the plates was replaced with the 10% AlamarBlue media and allowed to incubate for 4 h. The cell culture media was then assessed for fluorescence, (λ_ex_ = 570 nm, λ_em_ = 585 nm). Employing GraphPad software (Version 7.05) and using an Emax model, the mid-point of the inhibitory concentration curve (or IC_50_) was determined.

### 2.5. Effect of DcNP on Leukemic Cell Drug Uptake and Retention

One million HL-60 cells were aliquoted into several 1.5 mL Eppendorf tubes. To evaluate the effects of DcNP on VZ, we made a free-drug solution of venetoclax and zanubrutinib (1:1 *w/w*). A VZ-DcNP solution of identical drug concentrations was made and also added to each tube. Following drug exposure in a CO_2_ incubator, cells were removed at preselected timepoints (15 min, 40 min, 1 h, 1.5 h, 2 h, 3 h, and 4 h), and were then washed twice with media to remove external drug and VZ-DCNPs. Cells were then lysed with acetonitrile. Drugs in the cells were quantified according to the aforementioned extraction protocol and LC-MS/MS methods.

### 2.6. Pharmacokinetic Analysis of VZ-DcNP versus Free Drugs

All animal studies were performed under a protocol approved by the University of Washington Institutional Animal Care and Use Committee. Female BALB/c mice were purchased from Charles River Laboratories (Wilmington, MA, USA). They were housed in a pathogen-free facility until use with a 12 h light/dark cycle. Three groups of three mice each were tested as follows: (group 1) an IV dose containing 30 mg/kg venetoclax and 30 mg/kg zanubrutinib in 0.9% NaCl, 20 mM NaHCO_3_ buffer with 5% DMSO and 5% Cremophor EL as solubilizing agents; (group 2) an intravenous dosing of venetoclax and zanubrutinib DCNPs equivalent in volume and drug molar concentration to that received by group 1; and (group 3) a subcutaneous injection of venetoclax and zanubrutinib DCNPs in the inner thigh of the right back leg. Plasma samples were collected at 5 min, 1 h, 3.5 h, 24 h, 48 h, 72 h, and 1 week through retro-orbital bleeding. The drugs in plasma samples were extracted and analyzed with an HPLC-MS/MS.

## 3. Results

### 3.1. Design and Characterization of Nanoformulation and Production

#### 3.1.1. Effect of Solvent Removal, Size Reduction, and Drug/Lipid Ratio on Particle Size

To develop a venetoclax and zanubrutinib drug combination in a stable nanoparticle suitable for delivery to leukemia patients, biocompatible and biodegradable lipids were chosen for their nanoparticle structural ability as well as their ability to non-covalently associate with drugs across a wide range of hydrophilicity and hydrophobicity. The biocompatible lipids DSPC and DSPE-mPEG2000 were first mixed and dissolved together in organic solvent (either ethanol with 5% ammonia or tert-butyl alcohol); subsequently, the solvent was removed to form a lipid-drug complex. Using ethanol as the base solvent, the lipid-drug film was subjected to sonication as a particle size reduction process to form drug combination nanoparticles (diameter ~40 nm). This size range was chosen to enhance lymphocyte and leukemic cell uptake.

As outlined in Table 1, we first examined the process of solvent removal. Initially, we used rotary evaporation under reduced pressure for medium batches before later using lyophilization on a larger scale (>30 mL). We found that lyophilization did not seem to affect particle size or shape, but the process was retained to ensure complete removal of residual organic solvent.

We then determined whether varying ratios of drugs (venetoclax and zanubrutinib) to lipids (DSPC and DSPE-mPEG2000) influenced the particle size of DcNPs. Venetoclax and zanubrutinib were dissolved in a 1:1 molar ratio in small- and medium-sized batches as described above. The 1:1 ratio was selected due to its similarity to clinical treatments in human patients, while the drug amounts used were to approximate IC_50_’s of the drugs in a living system. Lipid mass was kept consistent throughout the experiment to assess the effect of drug mass on particle size, with the drug/lipid ratio ranging from 0 to 0.3. No effect on particle size was observed, so the selected drug concentrations were 3.63 mM venetoclax and 6.67 mM zanubrutinib with a drug/lipid ratio of 0.26 for later characterization and experimental use.

#### 3.1.2. Physical Characterization of VZ-DCNPs

To further characterize VZ-DCNPs, we evaluated the degree of VZ association to DcNP under sink conditions and visualized VZ-DCNPs using transmission electron microscopy. Following a 4-h dialysis to remove any unbound drug, we found that venetoclax drug association was nearly 100% and that of zanubrutinib was 98.5%, demonstrating a stable association of the drugs to DcNP. This strong drug association to DcNP may allow the nanoparticles to fulfill their purpose of synchronized drug delivery to cells.

As seen in Figure 1, VZ-DCNPs were visually examined using a transmission electron microscope, which revealed their consistent lozenge shape, with an average length of 39 ± 4 nm and an average width of 20 ± 2 nm. The nanoparticles appeared discrete and homogenous. The TEM image both supported our understanding of the nanoparticles and set a benchmark for future nanoparticle production.

### 3.2. Effect of DcNP on Venetoclax–Zanubrutinib Combination to Inhibit Leukemic Cell Growth 

To test the ability of VZ-DCNPs to kill cancer cells, three immortalized cell lines representing different types of leukemia were incubated with each test drug, or drugs in combination, for five days before measuring their relative growth. Respective drug effects are presented as the half-maximal inhibitory concentration, or IC_50_, and are summarized in Table 2. Free VZ combinations at equivalent concentrations and ratios were used as controls. HL-60 cells had the highest sensitivity to both free venetoclax alone (IC_50_: 1.92 ng/mL) and to the free combination drug 1:1 mass ratio with zanubrutinib (IC_50_: 0.181 ng/mL). HL-60 cells express both Bcl-2 and BTK, the respective targets of venetoclax and zanubrutinib, so their strong response to these drugs was expected. MOLT-4 and K-562 were less sensitive to venetoclax: 1.96 μg/mL and 15.9 μg/mL, respectively. This low response was also expected, as MOLT-4 cells express Bcl-2 at very low levels, and K-562 cells do not express the protein at all. Additionally, both MOLT-4 and K-562 were less sensitive to the free drug 1:1 mass ratio combination: 2.0 and 8.0 μg/mL, respectively. Zanubrutinib exhibited similar sensitivities for all cell lines tested; however, zanubrutinib is less potent with IC_50_s recorded for HL-60: 10.3 μg/mL, K-562: 8.3 μg/mL, and MOLT-4: 4.0 μg/mL. In contrast, the same set of drugs in DCNPs exhibited a much lower IC_50_ recorded value for HL-60 cells: 2.2 pg/mL (each at 1:1 *w/w* fixed ratio). These results demonstrate that VZ in DcNP remained active for their respective pharmacological targets and that they had enhanced potency to kill cancer cells when compared to free-drug equivalents.

### 3.3. Enhanced Uptake and Retention of Nanoparticle-Associated Drugs into Immortalized Leukemic Cell Lines

Cancer cell toxicity due to the nanoparticles (specifically, how the vehicle affects the associated drugs and their uptake) was then examined to assess the mechanism of toxicity in leukemia cell lines (HL-60, K-563, and MOLT-4). We found that both free drugs and those in DCNPs were rapidly taken up into all tested leukemic cells, reaching their peak intracellular drug concentrations within 1 h (Figure 2).

Intracellular venetoclax concentrations peaked at 1 h and were maintained for the 4-h study (terminal time point). Peak intracellular drug concentration was recorded at nearly 200 ng of drug per million cells (HL-60: 192 ng/million cells; K-562: 192 ng/million cells; MOLT-4: 176 ng/million cells) for cells exposed to the free-dosage form, compared to approximately 700 ng of drug per million cells for those incubated with an equivalent dose of DCNPs (HL-60: 674 ng/million cells; K-562: 647 ng/million cells; MOLT-4: 718 ng/million cells). These data suggested DcNP-associated venetoclax was taken up 3.5-fold compared to the uptake of the free-dosage form.

Intracellular zanubrutinib concentration also peaked at 1 h and persisted for the duration of the 4-h study. Cells incubated with the free drug reached an average maximal concentration of approximately 75 ng of drug per million cells (Hl-60: 69 ng/million cells; K-562: 109 ng/million cells; MOLT-4: 42 ng/million cells), compared to those treated with an equivalent dose of DcNP, which provided maximal values of 200–650 ng drug per million cells (HL-60: 256 ng/million cells; K-562: 647 ng/million cells; MOLT-4: 208 ng/million cells). Thus, DcNP-associated zanubrutinib was taken up 3-fold to 9-fold compared to free zanubrutinib. 

These in vitro cell-uptake kinetic data suggest that both the increased rate and extent of drug uptake due to nanoparticle association will positively affect the drugs’ ability to kill cancer cells. This was further explored through how the nanoparticle vehicle affected the pharmacokinetics of the associated drugs in vivo.

### 3.4. Effect of DcNP on Venetoclax and Zanubrutinib Pharmacokinetics in Mice

To investigate the effects of the DcNP nanoparticle on the pharmacokinetics of venetoclax and zanubrutinib, three groups of mice were intravenously administered with equivalent dosages of intravenous free drug, intravenous VZ-DCNPs, or subcutaneous VZ-DCNPs, all in 180 µL. Venetoclax was detectable in plasma for up to seven days for mice treated with the DcNP-dosage form, while those treated with an equivalent dose of free zanubrutinib were detectable for less than one day (Figure 3). Among the mice tested, those treated with subcutaneous VZ-DcNP had the highest extended plasma levels over the seven-day study (Table 3). This group of mice also exhibited the highest AUC values for both drugs: venetoclax in DcNP = 232 µg·mL^−1^·h (as compared to free drug = 88.8 µg·mL^−1^·h) and zanubrutinib in DcNP = 49 µg·mL^−1^·h (as compared to free drug = 8.3 µg·mL^−1^·h). Intravenously administered VZ-DCNPs had consistently higher venetoclax AUC’s than those treated with the free-drug counterpart. Venetoclax AUC in VZ-DcNP was 216 µg·mL^−1^·h (as compared to a free-drug AUC of 88.8 µg·mL^−1^·h) and zanubrutinib AUC was 11.3 µg·mL^−1^·h (as compared to a free-drug AUC of 8.3 µg·mL^−1^·h). It appeared that subcutaneously delivered VZ in DcNP provided the highest plasma drug exposure for both drugs. These results suggest that a subcutaneous administration of the VZ-DCNPs may be a safe and effective way to treat leukemia with infrequent administrations due to the long-acting plasma drug time-course extension of both venetoclax and zanubrutinib in combination.

## 4. Discussion

Taking advantage of our ability to co-formulate venetoclax and zanubrutinib in a drug combination nanoparticle (DcNP), we have characterized the VZ-DcNP as stable with a high degree of drug association to DcNP, indicating that further purification is not necessary (Table 2). In addition, the resulting VZ combination in DcNP is biologically active and shown to enhance the overall potency of VZ in fixed-dose combination at a 1:1 mole ratio (Table 3). The overall enhanced potency against HL-60 leukemic cells appeared to enhance cell uptake by 42- and 5-fold for V and Z, respectively. We found that VZ-DCNPs are stable, scalable, and biocompatible, as the lipid excipients, DSPC and DPSE-mPEG_2000_, provide a structural base to support the drug combination in nanoparticles appropriate for patient administration. These VZ-DcNPs are shown to extend plasma time-course and enhance the overall drug exposure of both venetoclax and zanubrutinib.

Unlike carriers that require drug encapsulation, it is noteworthy that both venetoclax and zanubrutinib were nearly completely associated to the DcNP lipid base structure in the VZ-DcNP. Thus, there is little or no drug loss during their synthesis, meaning that a final step to eliminate residual free drug is no longer necessary. In fact, our data indicate that the degree of drug association of both drugs to the VZ-DCNPs is 98% or more. Having a high degree of stable drug association may both reduce drug wastage in VZ-DcNP preparation and potentially minimize the risk of contamination while making the injectable VZ-DcNP product. In addition, we found that a lyophilized dosage form of VZ-DcNP could be produced, and that upon resuspension the product exhibited a mean diameter of ~30–40 nm, suitable for both intravenous and subcutaneous dosing. Coupled together, the high degree of association and the option of storing the drug product in lyophilized form (for producing a suspension on site) make VZ-DcNPs realistic for human testing.

While the exact mechanisms responsible for the enhanced potency of VZ-DcNP compared to that of its free VZ counterpart are still unclear, it is likely that the small nanoparticles promote leukemic cell uptake and retention. The DcNP-associated drugs are taken up faster and are maintained at higher concentrations compared to equivalent soluble (free) VZ (Figure 2). The enhanced uptake of VZ in DcNP parallels the improved potency of VZ in HL-60 leukemic cells expressing both VZ targets (Bcl-2 and BTK). This potency was also seen in K562 cells (BTK-expressing cell line) and MOLT-4 cells (Bcl-2-expressing cell line), indicating that both drugs were present and active. We found a roughly 82-fold enhanced potency for VZ-DcNP compared to the free formulation, and a 42- and 5-fold enhancement in cellular uptake for V and Z, respectively. Once again, the mechanisms leading to the disparity in the changes in cellular uptake are not clear, and they will be a subject of our future investigations. Regardless of the exact mechanisms, however, it is clear that both V and Z in DcNP are localized in the cells and are biologically active, leading to enhanced leukemic cell-growth suppression.

When delivered subcutaneously to BALB/c mice, VZ-DcNP was able to extend the presence of both drugs in plasma over an extended period of time, resulting in a significant extension of drug half-lives compared to those of free drugs given intravenously; a 42-fold increase for venetoclax and a 5-fold increase for zanubrutinib were observed. Intravenous VZ-DCNPs did not significantly alter the two drugs’ pharmacokinetics. VZ-DCNPs administered subcutaneously can greatly extend the plasma half-lives of associated drugs, demonstrating their ability to safely administer drugs over a longer period from a single injection, compared to the oral dosage forms that require more frequent dosing to achieve the same effect.

Currently, oral zanubrutinib (taken twice daily) and venetoclax (taken once daily) are in a phase 2 clinical study as a potential combination therapy for treating MCL (mantle cell lymphoma) and CLL (chronic lymphocytic leukemia) (NCT 05168930). Producing a long-acting and effective fixed-dose combination therapy intended to improve leukemic cell uptake and extend the time between doses may improve both patient uptake and acceptance.

New administration strategies for long-acting delivery of drugs that can overcome these limitations have been introduced. Long-acting cabotegravir with long-acting rilpivirine, explored through the CUSTOMIZE Hybrid III implementation-effectiveness study, is a novel formulation strategy that has been successfully implemented in HIV patient treatment, as the formulation can overcome the common problems with long-term drug treatment, namely patient adherence to the drug and maintaining adequate plasma drug levels. The VZ-DCNPs reported here can overcome these limitations imposed by daily oral dosing via subcutaneous administration of the nanoparticles: association with the biocompatible lipids safely retains the drugs in the subcutaneous space, protecting them from gastrointestinal and plasma metabolism while also slowly releasing the drugs over time into the plasma either through direct extravasation or through lymphatic uptake. These routes are likely responsible for the observed extended half-lives of the drugs, though more research is needed. 

In leukemic cells, a fixed-dose combination of venetoclax and zanubrutinib exhibited good potency, with IC_50′_s in the low nanogram per milliliter range for HL-60 cells, which have high expression levels of both drugs’ targets. When formulated as a free drug with the same fixed-dose combination as VZ-DCNPs, the IC_50_ value for HL-60 cells was enhanced by about 1000-fold. The improvement in potency of the DCNPs over the free-drug combination is likely due to enhanced uptake and retention of the DcNP-bound drug as compared to the free drug. The pharmacokinetic study results indicate that in mice, subcutaneously administered VZ-DcNP was more favorable than both intravenous VZ-DcNP and intravenous free drug. Both drugs were detectable for a longer period in the plasma of subcutaneous VZ-DcNP-treated mice those that received free drug or DcNP-associated drug through an intravenous injection.

## 5. Conclusions

Using lipid excipients to stabilize both venetoclax and zanubrutinib in drug combination nanoparticles, we successfully developed and characterized a VZ-DcNP injectable formulation that can be made in simple steps without the need for free-drug removal. VZ-DcNP is biologically active, is capable of enhancing leukemic cell uptake, and can extend both the drugs’ plasma half-lives and overall exposure per dose in mice. Both subcutaneous and intravenous administration of the nanoparticles provided larger overall exposure and drug half-lives than an equivalent intravenous administration of free drug, though subcutaneous administration provided the largest effect; in contrast, the effect from intravenous administration was minor. The VZ-DcNP fixed-dose drug combination may provide a long-acting pharmacokinetic profile and enhance overall drug exposure per dose; most importantly, it may synchronize the uptake and retention of both drugs in leukemic cells for a durable leukemia suppression, though these results remain preliminary and require further validation. We hope to further examine VZ-DcNP pharmacokinetics and safety in both mice and larger model organisms to explore the feasibility of VZ-DcNP as a future long-acting treatment for leukemia.

## 6. Patents

Patent disclosure has been made.

## Figures and Tables

**Figure 1 pharmaceutics-15-01016-f001:**
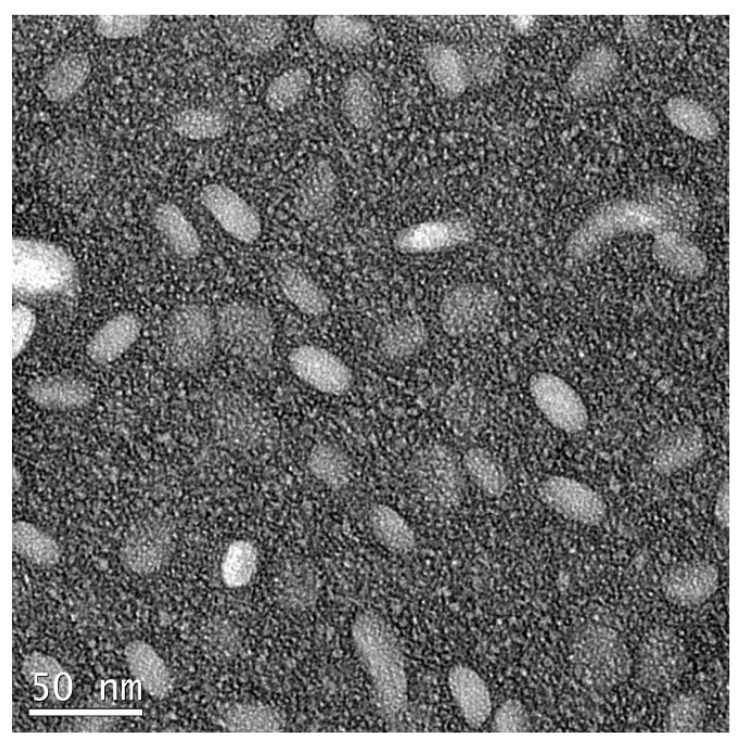
VZ-DCNPs examined with transmission electron microscopy with negative staining by 5% uranyl acetate. The drug combination nanoparticle, composed of the two drugs, venetoclax and zanubrutinib, appeared to assemble into solid, discrete, lozenge-shaped particles of approx. 20 nm × 39 nm with no apparent membrane structures.

**Figure 2 pharmaceutics-15-01016-f002:**
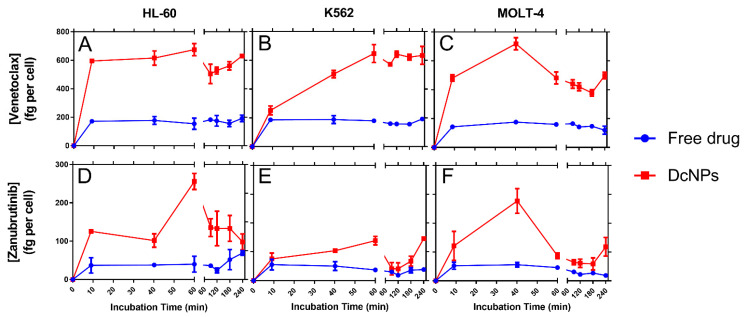
Effect of DcNP vehicle on rate and total uptake of venetoclax (**A**–**C**) and zanubrutinib (**D**–**F**) into leukemic cells. Drugs were incubated at a 1:1 mass ratio (3.63 mM venetoclax and 6.67 mM zanubrutinib) as either free drugs or nanoparticle-associated drugs.

**Figure 3 pharmaceutics-15-01016-f003:**
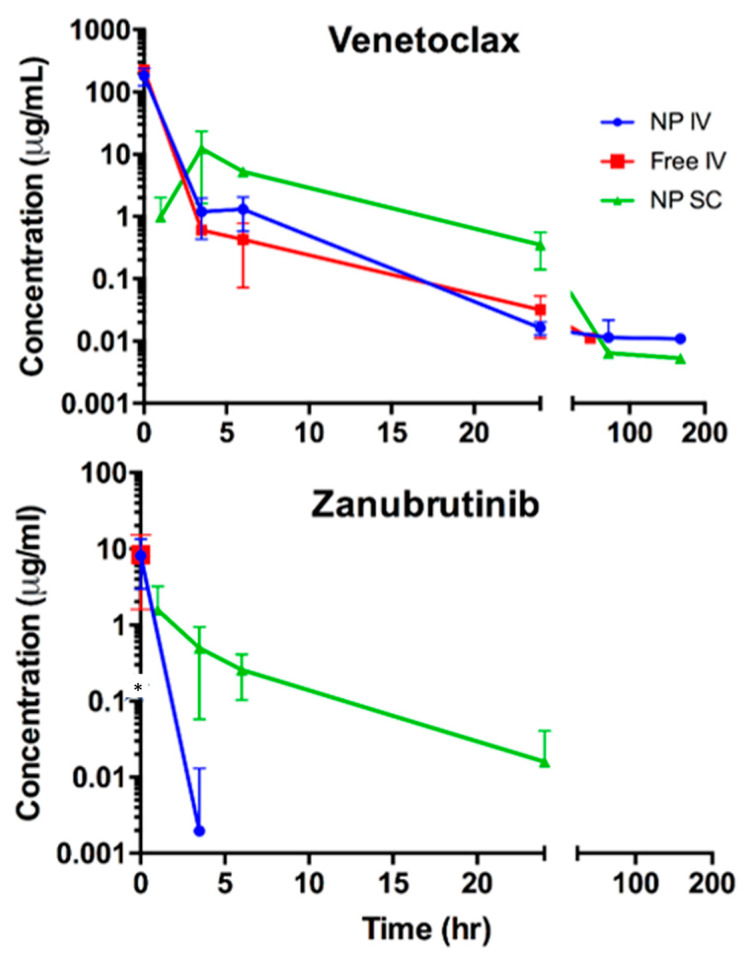
Effects of DcNP on pharmacokinetics of venetoclax and zanubrutinib. The venetoclax-zanubrutinib (1:1 *w/w*) nanoparticle (NP) combination was given by IV at 30 mg/kg each for route: NP-IV, NP-SC, or free form (Free-IV), respectively, to BALB/c mice. Plasma samples were collected at the indicated times; drug concentrations were determined over a 1-week study. Note: Drug concentrations below the limit of quantification were not plotted. * indicates an undetectable level of zanubrutinib following intravenous free administration.

**Table 1 pharmaceutics-15-01016-t001:** Effect of variations in solvent removal, particle size reduction, and drug/lipid ratios on nanoparticle size.

Preparation Method	Production Scale	Batch	Drug Concentration (mM)		Drug/Lipid Molar Ratio	Size (nm)
			Venetoclax	Zanubrutinib		
Solvent Evaporation > Hydration > Sonication	Medium	1	10	10	0.2	28
	(5–30 mL)	2	15	15	0.3	25
		3	5	5	0.1	11
		4	10	10	0.2	38
Lyophilization > Hydration	Large	1	3.63	6.67	0.26	30
	(>30 mL)	2	3.63	6.67	0.26	14

**Table 2 pharmaceutics-15-01016-t002:** Effects of VZ-DCNPs and free drugs (V, Z, or both) on leukemic cell growth inhibition.

	Cell Line	HL-60	K-562	MOLT-4
	Bcl-2 Expression	+	-	+
	BTK Expression	+	+	-
**Inhibitory Effects (IC_50_, pg/mL)**	Venetoclax	1.9 × 10^3^	1.6 × 10^7^	2.0 × 10^6^
	Zanubrutinib	1.0 × 10^7^	8.4 × 10^6^	4.0 × 10^6^
	V + Z Free Combo	180.6	8.0 × 10^6^	1.9 × 10^6^
	VZ-DcNP	2.2	-	-

**Table 3 pharmaceutics-15-01016-t003:** Effect of DcNP on plasma half-lives and exposures (AUCs) of venetoclax (V) and zanubrutinib (Z) in mice.

	Drug Route and Vehicle					
	Intravenous-Free		Intravenous-DcNP		Subcutaneous-DcNP	
	V	Z	V	Z	V	Z
t_1/2_ (h)	0.39	0.33	0.45	0.95	16.71	1.62
AUC (µg·mL^−1^·h)	400.6	15.1	338.7	14.2	98.9	7.2

## Data Availability

The data is available on request from the corresponding author. The data are not publicly available due to privacy of ongoing research.

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
