# Peer review of "Design and Characterization of a Novel Venetoclax-Zanubrutinib Nano-Combination for Enhancing Leukemic Cell Uptake and Long-Acting Plasma Exposure"

_pharmaceutics, 2023, doi:10.3390/pharmaceutics15031016_

Round 1
Reviewer 1 Report (New Reviewer)
The work was centered on developing a lipid-based drug combination nanoparticle systems for the controlled delivery of Venetoclax and Zanubrutinib. I consider this a novel delivery approach and look forward to seeing the work grow over time. Generally, the authors addressed this theme well and I would suggest the following amendments:
Abstract:
1. line 32 – please check “commination” (in drug-commination). It appears there is a typographical error here. Could it have been combination as already used earlier in the abstract.
Introduction:
2. line 39 – Check; after marrow cells. It seems it is meant to be a full stop/period and not semi-colon.
3. Consider rewording sentence in line 39-41 (i.e., Historically, treatments for ….)
4. Line 63 – 64 – Treatment potency instead of “potency of the treatment” could be more appropriate.
5. Line 72-73 – Check “fill fatigue”. It appears there is a typo here.
6. Line 88-91 – Gastrointestinal (GI) absorption… the sentence needs to be revised so the intended meaning becomes clear. The word “prior” is out of place in its current position; “levels” should be included after “sub-therapeutic drug plasma” so it reads subtherapeutic plasma levels.
Method:
7. Line 142. What do authors mean by the extent of venetoclax and zanubrutinib association with the lipid particle …. ? Could this be encapsulation or drug entrapment of some sort?
8. The inclusion of an experiment (using a characterization technique) that shows that the possibility of any physicochemical interaction between the two drugs or both drugs and carrier system would add value to the work.
Results:
9. Line 248-249 - I would suggest We then determined instead of “We next determine
10. Line 270 – 271 Reword this sentence.
11. Line 273 – double check usage of “longene” (in longene shape)
12. Check Line 397 – 398
Conclusion:
13. Line 445 “characterized would fit in better.
114. Since different routes of administration were explored for in injecting the DcNP (i.e., intravenous and subcutaneous), adding a sentence or two on findings to the conclusion could add some value.
Author Response
Please see the attachment

Reviewer 2 Report (New Reviewer)
I have reviewed the manuscript. This is a nicely performed study and a well written paper, which for my point of view, deserves publication. I have a small comment/suggestion.
1) In the conclusion part, the authors may further comment on the future use of this long-acting nanoformulation. What is planning for next? etc.
Round 2
Reviewer 1 Report (New Reviewer)
.
This manuscript is a resubmission of an earlier submission. The following is a list of the peer review reports and author responses from that submission.
Round 1
Reviewer 1 Report
The manuscript from Griffin et al. generated nanoparticle combining the 2 drugs Venetoclax and Zanubrutinib and showed that these VZ-DcNP could be taken up by leukemic cell lines and showed enhanced plasma exposure when injected in mice. Although the VZ-DcNP represents a very interesting therapeutic, the work is still in a very preliminary phase with limited data and has several flaws in its current form.
A major drawback is the chosen cell lines (and also mouse model) that do not represent the correct disease model that the authors mentioned. Venetoclax and BTK inhibitors such as Zanubrutinib are break-through drugs for the treatment of B cell leukemia such as CLL and MCL as the authors repeatedly mentioned. Treatment of CLL was referred as the main long-term aim of this study throughout the manuscript (abstract, key words, introduction, discussion, conclusion). However, the 3 chosen cell lines are either T cell leukemia or myeloid leukemia cells. Thus the data cannot be used to support the potential of using VZ-DcNP in B cell leukemia. I suggest that the author should use more leukemic B cell lines (which are dependent on BTK and BCL-2) to support the rationale of the study. The manuscript will also benefit from an expansion of introduction and discussion about the potential of Venetoclax and Zanubrutinib in myeloid leukemia, for example. In the mouse experiment, the authors should consider examining drug uptake and retention in the immune organs such as lymph nodes and bone marrow, as they are the main sites of leukemia/lymphoma cell proliferation.
Other comments that the authors may consider to improve the manuscript:
- A control treatment of NP without drug-load is missing in all experiments. This control is needed to clarify the possible effects of the NP (e.g. toxicity) on cells and mice.
- In figure 3, there are no error bar nor indication of the mouse number per group. I assume from this figure that only 1 mouse per group was treated, and if so, no conclusion can be drawn from such cohort.
- Citation of relevant references should be added at several places. For example, the expression levels of BTK and BCL-2 in the chosen cell lines should be cited, or even better, shown in an experiment.
Reviewer 2 Report
Specific comments on the manuscript.
The Authors developed a venetoclax and zanubrutinib drug combination in a stable nanoparticle (DcNP) formulation suitable for delivery to CLL patients and tested it in immortalized leukemic cells showing that the DcNP associate drugs were taken up faster and maintained higher concentrations compared to soluble VZ, leading to enhanced leukemic cell-growth suppression. When delivered subcutaneously to mice VZ-DcNP was able to extend the presence of both drugs in plasma over time, resulting in a significant extension of drug half-lives compared to free drug given intravenously.
This approach is very promising, but the reported data are too preliminary and should be clinically confirmed before entering therapeutic practice.
Reccomendation for revision.
1. Abstract. Line 12: chronic lymphocytic leukemia instead of the generic leukemia.
Line 12-13: " the difficulties in reaching and maintaining therapeutic drug concentrations in the target tissues and cells" are only partially the cause of resistance to therapy. This should be reported in the text..
Line 26-28 The results are interesting, but their efficacy in vivo is still missing.
Line 28-30: " The VZ DcNP's may be used to produce a long-acting formulation that allows for less frequent dosing, improve patient acceptance and outcomes in the treatment of leukemia". All these conclusions are not supproted by solid experimental data.
2. Introduction: Lines 35-38. CLL is a chronic leukemia, while MCL is a lymphoma. The first one is generally an indolent neoplasm, while in the second one, generally, prevails an aggressive clinical course. Therefore, these diseases are only partially comparable regarding their biology and treatment.
Line 48: CLL instead of leukemia
Line 57: "CLL or related leukemik cancers" Which ones are the "related leukemic cancers"?
Line 82-93: In spite of the pharmacokinetic problems reported by the Authors the combination of oral venetoclax and zanubrutinib showed good results in clinical studies. Therefore, they should make an effort to better support their criticisms.
3. Methods. Lines 171-179: " Three immortalized leukemic cell lines were incubated with drug or drug combinations: K-562 (CML cells), MOLT-4 (ALL origin) and HL-60 (APL origin). Cells were selected for their different protein expression of BTK, and Bcl-2". However all the three lines are not comparable to CLL or MCL cells. The Authors should better explain this apparent discrepancy.
4. Results: adequate.
5. Discussion: Line 379-381. This phrase is not well expressed.
Line 400-419: this part repeat concepts and statements already set out in Introduction. The Authors should make an effort tro discuss the real advantages of the VZ-DcNP, taking into account the good results of oral VZ in recent clinical trials.
6. Conclusions: Regarding the phrase " The VZ-DcNP may provide long-acting fixed- drug combination to synchronize delivery of both drugs to leukemic cells, for a maximum suppression of CLL, leukemia in general and other cancers" the available data are too preliminary for making this conclusion.
7. Tables and Figures: adequate